# Stable and Flexible Synaptic Transmission Controlled by the Active Zone Protein Interactions

**DOI:** 10.3390/ijms222111775

**Published:** 2021-10-29

**Authors:** Sumiko Mochida

**Affiliations:** Department of Physiology, Tokyo Medical University, Tokyo 160-8402, Japan; mochida@tokyo-med.ac.jp

**Keywords:** action potential, active zone, Ca^2+^ channels, homeostatic synaptic plasticity, presynaptic plasticity, presynaptic proteins, synaptic vesicle

## Abstract

An action potential triggers neurotransmitter release from synaptic vesicles docking to a specialized release site of the presynaptic plasma membrane, the active zone. The active zone is a highly organized structure with proteins that serves as a platform for synaptic vesicle exocytosis, mediated by SNAREs complex and Ca^2+^ sensor proteins, within a sub-millisecond opening of nearby Ca^2+^ channels with the membrane depolarization. In response to incoming neuronal signals, each active zone protein plays a role in the release-ready site replenishment with synaptic vesicles for sustainable synaptic transmission. The active zone release apparatus provides a possible link between neuronal activity and plasticity. This review summarizes the mostly physiological role of active zone protein interactions that control synaptic strength, presynaptic short-term plasticity, and homeostatic synaptic plasticity.

## 1. Introduction

Synaptic transmission is initiated by the fusion of neurotransmitter-containing synaptic vesicles (SVs) with the presynaptic plasma membrane. Presynaptic action potential (AP) triggers this process by opening voltage-gated Ca^2+^ (Ca_V_) channels in the plasma membrane composed of a neurotransmitter release site with scaffolding proteins [1,2]. Ca^2+^ sensor proteins expressed on the SV membrane cooperating with fusion machinery of soluble *N*-ethylmaleimide-sensitive-factor attachment receptor proteins (SNAREs) complex mediate the SV fusion and the release of neurotransmitters from fused SVs towards postsynaptic receptors [1,2]. The neurotransmitter release site, the active zone (AZ), is a highly organized structure composed of sophisticated protein machinery [1,2,3] that enables regulation of SV states, such as tethering, docking to the AZ plasma membrane, priming, fusion, and replenishment of the release sites with SVs after fusion.

This review introduces, at first, recent findings on temporal regulation of SV states within 100 ms of AP [4], current models for AZ proteins’ assembly [5,6,7], and their functions in the regulation of each SV state and SV localization [8]. Then, I further discuss Ca^2+^ dynamics-dependent replenishment of the neurotransmitter release site with release-ready SVs that involves multiple protein cascades, such as calmodulin-binding [9] and phosphorylation and dephosphorylation of AZ proteins [10,11]. These protein reactions underlie the presynaptic short-term plasticity [9,12,13] and homeostatic synaptic plasticity [14,15,16]. The AZ release apparatus provides a possible link between neuronal activity and sustainable synaptic transmission.

## 2. SV Dynamics in the AZ after AP

Synapses are intercellular junctions between a presynaptic neuron and a postsynaptic neuron—precisely opposed pre- and post-synaptic specializations that contain electron-dense material on their plasma membranes [1]. Ultrastructural studies have demonstrated that some SVs are in contact with the presynaptic plasma membrane in the AZ [17]. Freeze-fracture electron micrographs after a depolarizing stimulus showed that SV fusion occurs close to intramembrane particles [17], which are labeled with an antibody against Ca_V_ channels [18]. This AZ architecture allows millisecond excitation/synchronous neurotransmitter release coupling.

To characterize spatial and temporal organization of SV fusion sites following AP, the ‘zap-and-freeze’ method, generating an AP with a 1 ms depolarization pulse and following high-pressure freezing at defined time points, was recently developed by Watanabe and co-workers [4]. Applying this approach on mouse hippocampal neurons in culture, they succeeded morphologically to capture SV dynamics with a millisecond timescale, at 5, 8, 11, and 14 ms, triggered by the stimulus (Figure 1). In unstimulated synapses and in the presence of tetrodotoxin, an AP blocker reagent, 2% and 1% of the synaptic profiles exhibited exocytic pits in the AZ, respectively. At 5 ms, 18% of the synaptic profiles exhibited exocytic pits in the AZ. Synchronous fusion of multiple SVs occurs within 5 ms throughout a single AZ. Fused SVs collapse into the plasma membrane by 11 ms. From 5 to 11 ms, asynchronous fusion follows in the center of the AZ. Remarkably, during synchronous fusion, docked SVs across all synaptic profiles are reduced by ~40%, whereas SVs close to the membrane but not docked (between 6 and 10 nm) slightly increase—possibly undocked vesicles from the AZ but still associated with the release site. Such SVs are proposed to exist in a “loose state”, with SNAREs, synaptotagmin-1, a Ca^2+^ sensor, and Munc13, an AZ protein, still engaged [19]. During asynchronous fusion, docked vesicles are not further depleted despite ongoing fusion, suggesting that SVs are recruited during this process. At 14 ms, docked vesicles are fully restored to pre-stimulus levels with newly docked vesicles. This fast recovery is Ca^2+^-dependent and temporary and lasts for 100 ms or less. Newly docked vesicles undock or fuse within 100 ms, suggesting that the sequence of rapid redocking, and subsequent slow undocking, may underlie the facilitation of synaptic transmission. These snapshot images at defined time points demonstrated time-dependent millisecond SV dynamics, synchronous and asynchronous fusion, undocking, and docking, controlled by transient Ca^2+^ elevation with opening of Ca_V_ channels with membrane depolarization accompanying AP.

Watanabe and co-workers captured evoked SV dynamics at 5 ms, showing that synchronous fusion of multiple SVs occurs within 5 ms throughout a single AZ. This observation is consistent with evoked synaptic currents with delay <1 ms and a typical rise time of 1–3 ms [20].

Dynamically primed SV states for buildup of the molecular machinery that mediates SV fusion have been summarized by Neher and Brose [21], as follows: (1) loose tethering and docking of vesicles to release sites, forming the nucleus of SNARE complex assembly, (2) tightening of the complex by association of additional proteins, and partial SNARE complex zippering, and (3) Ca^2+^-triggered fusion. Corresponding timescales with high-frequency synaptic activity and concomitantly increased Ca^2+^ concentrations can proceed within: step (1) 10–50 ms, step (2) 1–5 ms, and step (3) 0.2–1 ms. These timescales estimated by electrophysiological studies are similar to but a bit faster than those of morphological observations induced by an AP [4], due to the influence of residual Ca^2+^ with high-frequency synaptic activity. Furthermore, a recent study proposes that vesicles transit sequentially from a loosely docked state to a tightly docked state before exocytosis, and Mover, an AZ protein, selectively controls the release probability of tight-state primed SVs in the mouse calyx of Held [22].

## 3. AZ Proteins and Their Complex

The AZ is a highly organized multi-protein structure serving as a platform for SV exocytosis mediated by the SNARE complex nearby Ca_V_ channels (Figure 2) [3,23]. This arrangement establishes the tight spatial organization required for fast synchronous SV fusion upon Ca^2+^ entry, and sets the synaptic strength [24]. As discussed in Section 2, AP-triggered Ca^2+^ elevation-dependent SV dynamics between docked and undocked states in the AZ is likely controlled by fusion machinery proteins, SNAREs, Ca^2+^-senser protein synaptotagmin-1, and AZ proteins [4]. Involvement of Munc13, as an AZ protein, in this SV dynamics is proposed by Watanabe and co-workers [4]. Munc13 underlies SV replenishment [19] and activity-dependent augmentation of SV pool size [8], suggesting its potential role in the presynaptic short-term plasticity.

(**a**)

(**b**)

### 3.1. Munc13

Munc13 is a component of the AZ proteins’ complex which includes RIM (Rab3-interacting molecules), RIM-BP (RIM-binding protein), Bassoon, Piccolo, Liprin-α, and CAST (cytomatrix at the AZ-associated structural protein)/ELKS [25,26,27,28,29,30,31,32]. These AZ proteins are all relatively large proteins with significant domain structures that interact with each other, forming a large macromolecular complex (Figure 2) [1]. Super-resolution imaging of Munc13 shows that Munc13-1 forms clusters that correlate with individual secretory sites within a single AZ at glutamatergic hippocampal synapses [7]. This lateral patterning was also observed in freeze-fracture immuno-electron microscopic experiments, in which Munc13 and Ca^2+^ channel clusters are detected within subareas of the AZ [33].

### 3.2. RIM, RIM-BP, and Others

The AZ of hippocampal excitatory synapses contains several clusters of the scaffold RIM, and these clusters mark individual release sites [6]. RIM mediates linkage of docked SVs and Ca_V_ channels at the release site [34] (Figure 2). Thus, RIM deletion reduces levels of Ca_V_2 channels [35], impairs SV docking, and slows down the rate of exocytosis [3]. In addition, RIM deletion reduces levels of Munc13-1 [36] that interacts with the RIM N-terminal tail. RIM interacts with other AZ proteins and the C-terminal sequences of Ca^2+^ channels [37,38] (Figure 2). A molecular complex consisting of RIM and C-terminal tails of the Ca^2+^ channels, that determines recruitment of Ca^2+^ channels to the AZ, includes RIM-BP [3] and CAST/ELKS [28]. RIM-BP also organizes Ca^2+^ channel topography at a fast auditory pathway central synapse [39]. Double knockout of RIM and ELKS induces loss of Munc13-1, Bassoon, Piccolo, RIM-BP2, and Ca_V_2 channels, disruption of AZ structure, and loss of vesicle docking, but not loss of Liprin-α [3]. Among AZ proteins, RIM and RIM-BP are essential for inclusion of Ca_V_ channels in the AZ, while RIM, RIM-BP, and CAST/ELKS may be essential for constitution of the AZ structure.

### 3.3. Bruchpilot and CAST/ELKS

Bruchpilot, a *Drosophila* ortholog of CAST/ELKS, is required for Ca^2+^ channel clustering and AZ structural integrity [40,41]. A super-resolution micrograph shows that Bruchpilot forms a single ring composing a T-bar per AZ, however, at AZs of the glutamate receptor-deficient neuromuscular junction synapse, it forms multiple rings, suggesting homeostatic presynaptic potentiation [14] (see Section 7). In accordance with this observation, CAST/ELKS has a potential function in the mammalian AZ [42]. CAST controls Ca^2+^ channel density [43] and AZ size [44,45]. Studying synaptic transmission of a cultured sympathetic neuron, we have demonstrated that disruption of CAST interaction with RIM or Bassoon impairs synaptic transmission [29]. At a central excitatory synapse, impairment of Bassoon slows down SV reloading [46]. These observations suggest that the CAST/ELKS complex including RIM and Bassoon plays fundamental roles in setting the synaptic strength in excitatory synapses. However, the function of ELKS differs between excitatory and inhibitory synapses, suggesting distinct roles in presynaptic assembly [5].

### 3.4. AZ Assembly Models

Interactions of AZ proteins have long been assessed for AZ assembly models (Figure 2) [1,3]. However, knockout of single AZ proteins, except for RIM and RIM-BP, which are essential for Ca^2+^ channel clustering [35,47], causes mild effects on the AZ structure. Thus, it is unlikely that individual protein interactions have a major role in assembly [5]. A new in vitro study proposed a model whereby AZ assembly relies on liquid–liquid phase separation principles [48], showing that RIM, RIM-BP, and Ca^2+^ channels form dense clusters on the supported lipid membrane bilayers via phase separation. Furthermore, small unilamellar vesicles (SUVs) and SVs from rat brains coat the surface of condensed liquid droplets formed by AZ proteins RIM, RIM-BP, and ELKS via phase separation [23]. Remarkably, SUV-coated RIM/RIM-BP condensates are encapsulated by synapsin/SUV condensates, forming two distinct SUV pools reminiscent of the reserve and tethered SV pools that exist in presynaptic boutons [23]. The SUV-coated RIM/RIM-BP condensates can cluster Ca^2+^ channels anchored on membranes. These studies reconstitute a presynaptic bouton-like structure mimicking the SV-tethered AZ with its one side attached to the presynaptic membrane and the other side connected to the synapsin-clustered SV condensates (Figure 2a). Individual interactions of AZ proteins are possibly less important, but instead, multiple low-affinity interactions possibly drive AZ formation [5], as suggested by possible models of droplet-like condensates’ formation with protein interactions in SV clustering [49] and postsynaptic density [50]. Low-affinity interaction or liquid–liquid phase separation can contribute to membrane anchoring [23,48].

### 3.5. AZ Assembly Stability and Degradation

Bassoon and Piccolo are two high molecular weight components of the AZ, with hypothesized roles in its assembly and structural maintenance. Bassoon and Piccolo are critical regulators of presynaptic ubiquitination and proteostasis. Loss of Bassoon and Piccolo leads to the aberrant degradation of multiple presynaptic proteins, culminating in synapse degeneration. This is mediated in part by the E3 ubiquitin ligase Siah1, an interacting partner of Bassoon and Piccolo whose activity is negatively regulated by their conserved zinc finger domains [51]. Inconsistently, the destruction of SVs in boutons lacking Piccolo and Bassoon was associated with the induction of presynaptic autophagy, a process that depended on poly-ubiquitination, but not the E3 ubiquitin ligase Siah1. Gain or loss of function of Bassoon alone suppressed or enhanced presynaptic autophagy respectively, implying a fundamental role for Bassoon in the local regulation of presynaptic autophagy. Mechanistically, Bassoon was found to interact with Atg5, an E3-like ligase essential for autophagy, and to inhibit the induction of autophagy in heterologous neurons. Loss of function of Atg5 as well as targeting an Atg5-binding peptide derived from Bassoon inhibited presynaptic autophagy in boutons lacking Piccolo and Bassoon [52]. Bassoon and Piccolo likely play a role in stabilizing the AZ by inhibiting degradation.

## 4. Control of SV States and Localization

### 4.1. SV States

The proteins that constitute the cytomatrix of the AZ play a central role in neurotransmitter release by localizing and clustering Ca^2+^ channels, and by regulating the states of the SV, such as tethering, docking, priming, and fusion (Table 1) [1]. Ultrastructural studies by Watanabe and co-workers have showed that AP triggers fusion of docked SVs, undocking of SVs, and redocking of SVs within 100 ms [4]. Undocked vesicles, between 6 and 10 nm from the AZ, are associated with the release site [4], and are proposed to exist in a “loose state” [21], engaged with SNAREs, synaptotagmin-1, and Munc13 [19]. Before AP firing, SVs are tethered and docked in the release site, and primed for fusion. RIM and RIM-BP, important proteins for forming condensed liquid droplets [48], as described in Section 3, play a role in SVs’ docking [38,39,53]. Knockout of individual proteins in mouse hippocampal neurons reduced docked vesicles [35,36], and double knockout of RIM and RIM-BP caused a loss of docked and tethered vesicles [54]. As described in Section 3, it was recently found that SVs from rat brains coat the surface of condensed liquid droplets formed by active zone proteins RIM, RIM-BP, and ELKS via phase separation, suggesting that SVs can be tethered to the surface of phase-separated AZ condensates [23]. Double knockout of RIM and ELKS also caused a loss of docked and tethered vesicles [3].

### 4.2. SV Docking and SV Priming

SVs from rat brains coat the surface of condensed liquid droplets formed by AZ proteins RIM, RIM-BP, and ELKS via phase separation [23], suggesting that ELKS/CAST also determine SV pool size in the AZ. Overexpression of CAST in cultured sympathetic presynaptic neurons with microinjection of the recombinant DNA increased the size of the AZ, to 4-fold greater than the normal AZ [10]. Indeed, CAST overexpression prolonged the duration of the excitatory postsynaptic potential (EPSP) and increased the integral by 1.7-fold, due to an increase in the number of SVs in the readily releasable pool (RRP) [58], whereas the EPSP peak amplitude was unchanged. These results suggest that CAST controls the AZ size and the number of docked SVs, but not the SV release probability or SV priming. However, ELKS enhances the release probability at inhibitory hippocampal synapses [53,59], while it enhances the RRP at excitatory synapses without affecting the release probability [70]. ELKS C-terminal sequences, which interact with RIM, are dispensable for the RRP enhancement [70]. Instead, the N-terminal ELKS coiled-coil domains that bind to Liprin-α and Bassoon are necessary to control the RRP [70]. Thus, ELKS has differential, synapse-specific effects on the RRP (SV docking) and the release probability (SV priming), although it is proposed that ELKS N-terminal domains have important roles in SV priming [70].

RIM mediates SV docking and priming, while Munc13 forwards SVs to the primed state [36], leading to a SV in the RRP that can rapidly fuse upon Ca^2+^ stimulation [60]. Munc13-1 together with Munc18-1 stabilize release-ready vesicles by preventing NSF-dependent de-priming [65]. Munc13s play a key role in determining the number of release sites and fusion-competent vesicles [7,61]. As conserved calmodulin binding sites in Munc13s, which are essential regulators of SV priming and synaptic efficacy, Ca^2+^ sensor/effector complexes consisting of calmodulin and Munc13s regulate SV priming and synaptic efficacy in response to a residual Ca^2+^ signal, and thus shape short-term plasticity characteristics during periods of sustained synaptic activity [9].Molecular perturbation studies in hippocampal neurons suggest that RIM-BPs promote SV priming by interaction with Munc13 [55].

As described above, RIM, interacting with cytoplasmic C-terminal sequences of Ca^2+^ channels [37,38] and SVs via Rab3 [69], may participate in the control of release probability with Ca^2+^ elevation, while Munc13 may act as a modulator of SVs for determining the SV pool size in the AZ. During repeated APs, Ca^2+^ and phosphatidylinositolphosphate (PIP) binding to the Munc13 C2B domain potentiate SV exocytosis, thereby offsetting synaptic depression induced by vesicle depletion [71]. A possible Ca^2+^-independent action of RIM for control of release efficacy was also proposed, as follows: the Ca^2+^-independent interaction between RIM C2B domains and PIP2 coordinates the efficient function of fusion machinery by controlling the relative position of PIP2 and secretory proteins that depend on it [64].

### 4.3. SV Fusion

For SV fusion, before the actions of the Ca^2+^ sensor and SNAREs fusion machinery, Munc18 accelerates SNARE proteins assembling through direct contact with both t- and v-SNAREs [72]. Deletion of Munc18-1 in mice leads to a complete loss of neurotransmitter secretion from SVs throughout development [73]. However, this does not prevent normal synaptic assembly and formation of morphologically defined synapses, suggesting that synaptic connectivity is independent of neurotransmitter secretion [73]. Munc13-1 has been shown to promote membrane fusion: it tethers SVs to the plasma membrane through its N- and C-terminal C_2_ and C_1_ domains [66,74] and directly enhances SNARE assembly through the MUN domain [62,75,76]. Munc13-1 was shown to cooperate with Munc18-1 to promote the accuracy of SNARE assembly [62]. Munc13-1 and Munc18-1 cooperatively chaperone SNARE assembly [77]. Thus, Munc18-1 and Munc13-1 together serve as a functional template to orchestrate SNARE complex assembly [76].

### 4.4. SV Localization

For the SV localization factor, the dynamic tripartite complex of α-RIMs, Munc13, and Rab3, an SV protein, is suggested to mediate SV targeting to the release sites [69]. Their roles in determining SV localization have been suggested with ultrastructural studies: cryo-electron tomographic experiments revealed fewer tethered vesicles and a lower vesicle density in RIM1α knockout synaptosomes, suggesting a distinct contribution of RIM in SV localization close to the AZ membrane [56]. Rosenmund and his co-workers have recently provided comparative ultrastructural evidence, as well as the comparative analysis of synaptic transmission with electrophysiology, in RIM and Munc13-deficient hippocampal synapses for dissecting the Munc13-dependent and -independent roles of RIM in SV localization and in regulation of SV release probability [8]. RIM1/2 localizes SVs in the proximity (1–20 nm) of the AZ membrane independent of Munc13-1, while both RIM1/2 and Munc13-1 mediate SV docking. To compare the roles of RIM and Munc13-1 in SV priming activity, the RRP of SVs was estimated by measuring the postsynaptic charge evoked by the hypertonic sucrose application, which induces Ca^2+^-independent neurotransmitter release from SVs in glutamatergic hippocampal autaptic neurons. Compared with the control neurons, the RRP size was severely reduced by Munc13-1 deletion (~95%) or RIM1/2 deletion (~88%), suggesting the requirement of both RIM1/2 and Munc13-1 for efficient SV priming. In addition, the activity of both RIM1/2 and Munc13-1 is essential for AP- or Ca^2+^-evoked SV fusion in the hippocampal autaptic neuron. Although neurotransmitter release efficiency is controlled uniquely by RIM1/2, activity-dependent augmentation of SV pool size relies exclusively on the action of Munc13s. Rosenmund and his co-workers concluded that RIM and Munc13-1 act in overlapping and independent states of SV localization and release. It is likely that, in the resting state, RIM localizes SVs in the right position, in the proximity of the AZ membrane, and then RIM together with Munc13 forward the SV state to docking and priming on the AZ membrane [56].

## 5. SV Replenishment

### 5.1. CAST/ELKS

Immunocytochemical experimentally, an activity-dependent increase in phosphorylated CAST^S45^ in sympathetic presynaptic terminals was observed with APs burst (20 < 50 Hz) [10]. The phosphomimetic-CAST^S45D^ expression in the sympathetic presynaptic neuron reduced the peak amplitude and the integral of EPSP, and the SV number in the RRP [10]. Thus, the CAST^S45^ phosphorylation reduces the number of release-ready SVs. The paired-AP protocol experiments indicate that phosphorylation of CAST^S45^ with the time window of 30–120 ms after the first AP induces paired-EPSP depression. Overexpression of the phosphonegative-CAST^S45A^-mutant reduces the paired-EPSP depression (<200 ms), suggesting that phosphorylated CAST^S45^ downregulates SV reloading shortly after AP, but not over longer (200 ms) time courses. This means that AP activates a kinase in the AZ to phosphorylate CAST^S45^ that negatively controls replenishment of the release site with release-ready SV. The possible kinase is an AZ-associated serine/threonine kinase SAD-B that phosphorylates CAST^S45^ in vitro [10].

Acute deletion of CAST by short-hairpin RNAs in sympathetic neuron synapses did not alter EPSP amplitude, however, it significantly delayed the rate of the fast reloading of release-ready SVs, following the RRP depletion with APs burst [10]. Indeed, CAST deletion facilitated the paired-EPSP depression [10]. These results indicate that CAST is required for the fast SV reloading, however, the phosphorylated CAST, <200 ms after AP, brakes transmitter release for the next arriving AP at the presynaptic terminal by slowing down the SV reloading. This presynaptic regulation is reasonable for a relatively low firing rate and slow signal conduction of sympathetic postsynaptic neurons. Bassoon, a binding partner of CAST/ELKS [29], is involved in the reloading of SVs to release sites at excitatory synapses [46]. Another binding partner, RIM1, interacting with Munc13-1, has been implicated in SV docking and priming [8]. These protein interactions are likely involved in the CAST-mediated fast replenishment of release sites with release-ready SV.

### 5.2. Bassoon

Bassoon mediates reloading of SVs to release sites at central excitatory synapses—cerebellar mossy fiber to granule cell synapses [46]. Bassoon deletion did not affect basal synaptic transmission, but it enhanced short-term synaptic depression during sustained high-frequency AP. The rate of SV reloading was halved in the absence of Bassoon. A slowing down of the SV replenishment rate at the central endbulb synapse in Bassoon deletion mice, where number and vesicle complement are normal, was also reported [16]. Thus, Bassoon speeds up activity-dependent SV tethering, resulting in rapid replenishment of the release site. At the rat calyx of the Held synapse, Bassoon and Piccolo, which are highly homologous scaffolding proteins, separately or simultaneously share functions in SV replenishment during high-frequency synaptic activity [52]. Furthermore, in Piccolo-deficient calyces, recruitment of slowly releasing SVs of the RRP, normally not visible for AP-induced release, can be observed during high-frequency stimulation, suggesting a specific role for Piccolo in organizing a sub-pool of the RRP that prevents depletion of release-ready SVs during prolonged and intense firing activity [52]. Additive roles of Piccolo and Bassoon in SV replenishment for control of the RRP size are reported in the fast central auditory synapse. There, Piccolo is unlikely to influence the release probability, while Bassoon seems to be a regulator of it [63].

### 5.3. RIM-BP

RIM-BP, binding to both RIM and Ca_V_ channels [47], is an essential organizer of the topography of SV release sites, as discussed in Section 3. Studies of the *Drosophila rbp* null mutation proposed that RIM-BP action participates in a rate-limiting stage required for the replenishment of high release probability SVs following vesicle depletion [57]. Recent findings in the fast central endbulb synapse of auditory nerve fibers to bushy cells of the cochlear nucleus also demonstrated that RIM-BP controls both the release probability and the SV replenishment [39]. RIM-BP2 deletion did not change the size of the RRP, but lowered the release probability, due to a slowed down rate of the Ca^2+^-dependent fast SV replenishment. Ultrastructural studies showed a reduction of docked and proximal SVs, in addition to an impaired topography of Ca_V_ channels in the AZ [39]. Binding to RIM-BP, Bassoon also controls Ca_V_2.1 channels’ localization in the AZ [67]. A genetic deletion of Bassoon and an acute interference with Bassoon and RIM-BP interaction reduce synaptic abundance of Ca_V_2.1 channels, weaken P/Q-type Ca^2+^ current-driven synaptic transmission of cultured hippocampal neurons, and result in a higher relative contribution of neurotransmission dependent on Ca_V_2.2 channels [67]. It is likely that RIM-BP, interacting with Bassoon and CAST/ELKS via RIM, modulates the rate of Ca^2+^-dependent fast SV replenishment.

## 6. Presynaptic Short-Term Plasticity

Short-term synaptic plasticity occurs during and after repetitive presynaptic AP activity on a timescale of milliseconds to minutes, and manifests in the modulation of synaptic efficacy via changes in the release probability and the RRP size [78].

### 6.1. RIM

RIM1, an essential protein forming droplet-like condensates which reconstitutes the SV-tethered AZ [23,48] (see Section 3), functions in presynaptic short- and long-term synaptic plasticity depending on different synapses [79,80]. At cerebellar parallel-fiber synapses, deletion of RIM1α reduces the release probability and consequently enhances the short-term facilitation, however, the long-term plasticity is fully intact [80]. At CA1-region Schaffer-collateral excitatory synapses and in GABAergic synapses, RIM1α is required for the short-term plasticity. In contrast, in excitatory CA3-region mossy fiber synapses and cerebellar parallel fiber synapses, RIM1α is required for the presynaptic long-term, but not short-term, plasticity. This long-term plasticity depends, at least in part, on phosphorylation of RIM1α at a single site, suggesting that RIM1α constitutes a ‘phosphoswitch’ that determines synaptic strength [79]. RIM-BP, another essential protein forming droplet-like condensates reconstituting the SV-tethered AZ [23,48], controls the rate of the Ca^2+^-dependent fast SV replenishment [39], suggesting its involvement in short-term plasticity.

### 6.2. Munc13 and Calmodulin

The neurotransmitter release probability is controlled by RIM, while activity-dependent augmentation of SV pool size relies on the action of Munc13 proteins [8] that function in SV priming [55,60]. Thus, Munc13 proteins’ contribution to the presynaptic short-term plasticity has been proposed [9,19]. Munc13-1 and the splice isoform ubMunc13-2 bind calmodulin in a Ca^2+^-dependent manner via an evolutionally conserved calmodulin recognition motif. Calmodulin binding to Munc13 proteins causes increased priming activity and the RRP sizes in autaptic hippocampal neurons. Thus, calmodulin and Munc13 form a Ca^2+^ sensor/effector complex and its activation by elevation of residual Ca^2+^ during and after repetitive AP shape short-term plasticity characteristics [9]. In mice calyx of the Held synapses, expression of Ca^2+^-calmodulin-insensitive Munc13-1 exhibits a slower rate of SV replenishment, aberrant short-term depression, and reduced recovery from synaptic depression after high-frequency stimulation. Therefore, Ca^2+^-calmodulin/Munc13-1 complex is a pivotal component of the molecular machinery that determines short-term synaptic plasticity characteristics [19]. Furthermore, in the presynaptic terminal of cultured sympathetic neurons, after AP generation, elevation in the Ca^2+^ level activates calmodulin, which in turn liberates Munc18-1, causing short-term synaptic facilitation, such as paired-pulse facilitation [81]. In the rat calyx of the Held presynaptic terminals, calmodulin mediates rapid recruitment of fast-releasing SVs that supports recovery from synaptic depression during high-frequency trains in concert with rapid recovery of the slowly releasing vesicles [68]. AZ proteins, such as Munc13, Munc18, and possibly others, are targets of Ca^2+^-calmodulin during and after repetitive AP to facilitate transmitter release.

### 6.3. Munc18, Diacylglycerol, and PKC

Activation of Munc13 by phorbol esters is essential for synaptic potentiation [82]. Protein kinase C (PKC) is a major intracellular target of Diacylglycerol. However, the Munc13-mediated synaptic potentiation in hippocampal autaptic neurons is PKC-independent [12]. In contrast, Munc18-1 mediates the synaptic potentiation downstream target in the PKC pathway. Expression of a PKC-insensitive Munc18-1 mutant in Munc18-1 null neurons does not show Diacylglycerol-induced potentiation and increased synaptic depression [12]. Post-tetanic potentiation (PTP) is a form of short-term plasticity that decays within tens to hundreds of seconds [78]. A recent “flash and freeze” electron microscopy study of hippocampal mossy fiber-CA3 pyramidal neuron synapses revealed that PTP is associated with an increase in the docked SV pool, forming structural ‘pool engrams’ [83]. The calyx of the Held synapse expresses robust PTP [13]. For PTP production, two PKC phosphorylation sites of Munc18-1 are critically important. A dynamic PKC phosphorylation/dephosphorylation cycle of Munc18-1 drives short-term enhancement of transmitter release during PTP [13]. On the contrary, at the hippocampal CA3-to-CA1 synapse and the granule cell parallel fiber (PF)-to-Purkinje cell (PC) synapse of a knock-in mouse, in which all Munc18-1 PKC phosphorylation sites were eliminated, 70% of PTP remained at CA3-to-CA1 synapses, and the amplitude of PTP was not reduced at PF-to-PC synapses [84]. These findings indicate that Munc18-1, dependently and independent of its phosphorylation by PKC, can have different effects on PTP production in different synapses.

### 6.4. Bassoon and Piccolo

Short-term depression during high AP activity, but not basal SV release, is controlled by Bassoon [46] and Piccolo [52]. The cerebellar mossy fiber to granule cell synapse in Bassoon knockout mice showed enhanced short-term synaptic depression during sustained high-frequency trains due to the reduced SV reloading rate in the absence of Bassoon [46]. At the rat calyx of the Held synapse with knockdown (KD) of Bassoon and Piccolo—highly homologous scaffolding proteins—separately or simultaneously, short-term depression was prominently increased by the Bassoon KD, whereas KD of Piccolo only had a minor effect [52]. Piccolo seems to have a function in the recruitment of slowly releasing SVs of the RRP, which are normally not available for AP-induced release, during high-frequency stimulation. Thus, the regulation of SV refilling during ongoing synaptic activity is a shared function of Bassoon and Piccolo, although Bassoon appears to be more efficient [52]. So far, no evidence for Ca^2+^-dependent activation of Bassoon and Piccolo for their regulation of the short-term depression has been reported, however, contribution of a kinase associated with the AZ cytomatrix is a possible candidate.

### 6.5. CAST and SAD-Kinase

In this section, the involvement of RIM1, Munc13, Munc18, Bassoon, and Piccolo in short-term synaptic plasticity during sustained high-frequency presynaptic AP activity is described. Lastly, for milliseconds duration presynaptic plasticity, I would like to introduce again a role of CAST in the paired-pulse depression in cultured sympathetic neuron synapses [10] (see Section 5). The second EPSP evoked shortly after the first AP is smaller than the first EPSP and recovers within 200 ms [85,86]. This is because of synaptic depression due to undocking and new docking of SVs at the release site after synchronous SV fusion [4]. SAD-kinase, a presynaptic kinase associated with the AZ cytomatrix and SVs, regulates neurotransmitter release [87], and in vitro phosphorylates CAST^S45^ [10]. Thus, SAD-kinase activated by an AP possibly phosphorylates CAST^S45^ in vivo. Expression of phosphonegative-CAST^S45A^ reduced the depression, while that of phosphomimetic-CAST^S45D^ increased the depression of the second EPSP even at 2000 ms after the first AP [10]. This means that CAST^S45^ phosphorylation potentiates paired-pulse depression, while dephosphorylation of CAST^S45^ reduces it. Therefore, phosphorylation of CAST^S45^, within 200 ms of AP firing, negatively controls rapid replenishment of the release site (see**Section 5**) with SV docking (see Section 4), resulting in paired-pulse depression.

## 7. Presynaptic Homeostasis Plasticity

As described in Section 3, Bruchpilot, a *Drosophila* ortholog of CAST/ELKS, required for Ca^2+^ channel clustering and AZ structural integrity [40,41], has a role in homeostatic presynaptic potentiation. Bruchpilot normally forms a single ring composing a T-bar per AZ, however, at AZs of the glutamate receptor-deficient neuromuscular junction synapse, it forms multiple rings [14]. Synaptic connections undergo homeostatic readjustment in response to changes of synaptic activity, to ensure a stable and flexible nervous system. Homeostatic signaling acts in both the central and peripheral nervous systems in various species, ranging from *Drosophila* to humans [14,88,89,90]. For example, at endplates of myasthenia gravis patients, to maintain muscle excitation, reduced postsynaptic sensitivity is counteracted by upregulated neurotransmitter release to restore evoked postsynaptic current amplitudes [91]. Thus, AZ proteins which regulate states and localization of SVs at the release site could functionally participate in the homeostatic regulation system. In this section, I will introduce roles of AZ proteins, the so far reported RIM, RIM-BP, and Bassoon, in presynaptic homeostatic plasticity.

### 7.1. RIM

At the *Drosophila* neuromuscular junction, in accordance with the effects of RIM deletion on mammalian presynaptic function, loss of RIM disrupts baseline neurotransmitter release, diminishes presynaptic Ca^2+^ influx, and decreases the RRP size [15]. In addition, loss of RIM blocks the homeostatic enhancement of neurotransmitter release [15]. This occurs after inhibition of postsynaptic glutamate receptors, a process termed synaptic homeostasis. Synaptic homeostasis requires enhanced presynaptic Ca^2+^ influx to potentiate SV release [15,92]. However, in *rim* mutants, the homeostatic modulation of Ca^2+^ influx proceeds normally, despite a defect in baseline Ca^2+^ influx [15]. Synaptic homeostasis is also correlated with an increase in the RRP size [15,92]. In *rim* mutants, the homeostatic enlargement of the RRP is blocked [15]. Thus, during homeostatic synaptic plasticity, RIM functions in the RRP enlargement but not in Ca^2+^ influx enhancement. For the homeostatic modulation of presynaptic Ca^2+^ influx, expression of the Degenerin/Epithelial Sodium channel (sodium leak channel), which modulates presynaptic membrane voltage and, thereby, controls Ca^2+^ channel activity for homeostatic neurotransmitter release, is required at the *Drosophila* neuromuscular junction [93].

### 7.2. RIM-BP

At the *Drosophila* neuromuscular junction, RIM-BP is also essential for the homeostatic modulation of neurotransmitter release, being required for both the enhancement of Ca^2+^ influx and the enlargement of the RRP [57]. In addition, RIM-BP is essential for the normal resupply of high release probability SVs. This function is independent of the modulation of Ca^2+^ influx and the RRP size. Loss of RIM-BP slows the rate of high release probability SV resupply by ~4-fold at basal synaptic activity [57]. In contrast, during homeostatic signaling, loss of RIM-BP slows the rate by around 7-fold [57]. It is possible that synaptic homeostasis acts upon a presynaptic vesicle pool of high release probability SVs residing near presynaptic Ca^2+^ channels and RIM-BP in the AZ. Thus, RIM-BP is required for stabilized baseline neurotransmission, coupling SVs to sites of Ca^2+^ influx, and for the persistent homeostatic modulation of presynaptic neurotransmitter release through coordinated control of Ca^2+^ influx and dynamics of the high release probability SV pool [57]. The high release probability SV is likely the primed SV in mammalian synapses (see Section 4). Molecular perturbation studies in hippocampal neurons suggest that RIM-BPs promote SV priming by interaction with Munc13 [55].

### 7.3. Bassoon

In a mammalian central nervous system, the fast central endbulb synapse of auditory nerve fibers to bushy cells of the cochlear nucleus, disruption of Bassoon slows SV replenishment and induces homeostatic plasticity [16]. In the Bassoon-deleted mice endbulb synapses, the number and complement of SVs in the AZs are normal. In contrast, postsynaptic densities, quantal size, and vesicular release probability are increased, while SV replenishment and the RRP are reduced. These various effects caused by Bassoon disruption integrate each other into an evoked EPSP (EPSC) with unaffected amplitude [16]. These findings indicate that presynaptic disfunction drives homeostatic plasticity both in the presynaptic and postsynaptic functions for synaptic upscaling. For Bassoon in the endbulb presynaptic terminal, it promotes SV replenishment to the release sites, and consequently, enlarges a pool size of readily releasable SVs (RRP) [16]. Bassoon mediates SV replenishment of release sites at central excitatory synapses—cerebellar mossy fiber to granule cell synapse [46] and the calyx of the Held synapse during high-frequency synaptic activity [52]. Thus, it is likely that Bassoon has a fundamental role in contribution to the synaptic homeostasis in the central nervous system.

## 8. Conclusions

At the presynaptic terminal, SV fusion takes place within the sub-millisecond opening of nearby Ca^2+^ channels, with membrane depolarization accompanying AP [20]. AP triggers not only neurotransmitter release from the fused SVs but also SV dynamics in the AZ, following millisecond Ca^2+^ dynamics that control SV states, synchronous and asynchronous fusion, undocking, redocking, and priming [4]. The undocking, redocking, and priming SV states contribute to presynaptic short-term plasticity [4]. Our electrophysiological study showed that AP-induced millisecond Ca^2+^ dynamics activates multiple protein cascades via Ca^2+^-sensor molecules [86,94] to control replenishment of the release site with release-ready SVs [95]. The AZ proteins complex, activated by the AP-induced millisecond Ca^2+^ dynamics, mediates the SV dynamics for regulation of neurotransmitter release probability, controlling the SVs/Ca^2+^ channel coupling. For this control, RIM and RIM-BP, required for clustering the Ca^2+^ channel at release sites [37,38,42], are major key proteins. The AP-induced millisecond Ca^2+^ dynamics also regulates the RRP size, controlling the SV states of docking and priming. CAST/ELKS and Munc13 are major key proteins for docking and priming, respectively [8,10,61]. Notably, ELKS has differential, synapse-specific effects on the RRP and the release probability, although it is proposed that ELKS N-terminal domains have important roles in SV priming [70].

During and after repetitive AP firing*,* SV replenishment supports stable synaptic transmission. Bassoon, Piccolo, and Clarinet [96], a homology with piccolo expressed in *Caenorhabditis elegans*, play a role in SV tethering under sustained high AP activity to maintain stable synaptic transmission [16,49,59].

Presynaptic plasticity is dependent on flexible neurotransmitter release. Presynaptic depression is due to reduced neurotransmitter release, while presynaptic facilitation or potentiation is due to increased neurotransmitter release. Residual Ca^2+^-dependent regulation of the release probability via SV fusion machinery has been proposed for the generation of presynaptic plasticity [78,97]. However, as reviewed in this article, the contribution of AZ proteins to presynaptic plasticity is significant. Modulation of RIM and RIM-BP function alters Ca^2+^ channel clustering nearby the release apparatus [37,38,42]. In addition, modulation of Munc18 may alter SV fusion speed [72], that of Munc13 and CAST/ELKS function alters the RRP size [8,9,10,70], and that of Bassoon and Piccolo alters SV tethering under repetitive AP firing [16,49,59]. Ca^2+^-dependent modulation of AZ protein function is mediated by calmodulin-binding [9,19,85], phosphorylation with protein kinases such as PKC [12,13] and SAD-kinase [10], and dephosphorylation [10]. Calmodulin and PKC mediate synaptic facilitation, while SAD-kinase mediates synaptic depression. Ca^2+^-independent modulation of AZ protein function also contributes to presynaptic plasticity [12,81,87].

Synaptic connections undergo homeostatic readjustment in response to changes of synaptic activity, to ensure a stable and flexible nervous system [14,88,89,90,91]. Important roles of RIM [15], RIM-BP [57], and Bassoon [16] in presynaptic homeostatic plasticity, that controls the release probability and the RRP, have been reported so far. Stable and flexible homeostatic synaptic transmission for non-stop signaling is likely supported by coordinated functions of RIM, RIM-BP, Bassoon, and other AZ proteins.

## Figures and Tables

**Figure 1 ijms-22-11775-f001:**
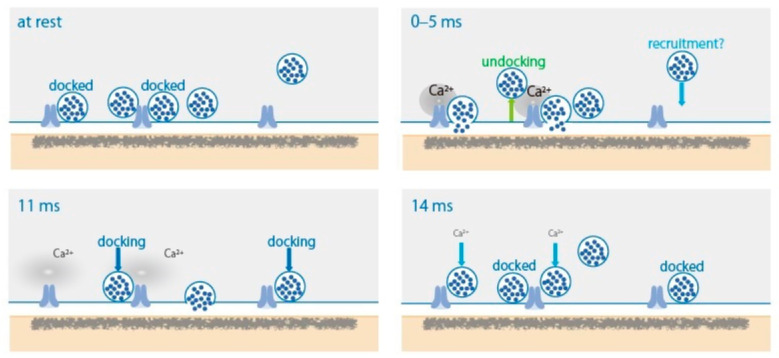
Diagram of synaptic vesicle dynamics in the active zone after a single action potential firing. (At rest) Vesicles close to the active zone are proposed to transit between docked and undocked states, resulting in a certain number of vesicles docked and ready to fuse in response to action potential firing. (0–5 ms) Action potential firing triggers synchronous fusion, often of multiple vesicles, begins within hundreds of microseconds, and the vesicles finish collapsing into the plasma membrane by 11 ms. From 5 to 11 ms, residual calcium triggers asynchronous fusion, toward the center of the active zone. (11 ms) New vesicles start to be recruited in less than 10 ms. (14 ms) New docked vesicles have fully replaced the vesicles used for fusion. These vesicles then undock or fuse within 100 ms. Reproduced from Kusick et al., 2020 [4].

**Figure 2 ijms-22-11775-f002:**
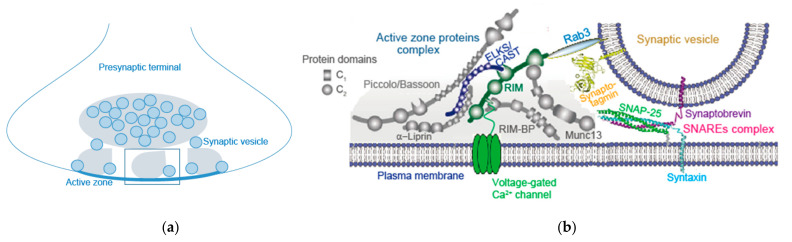
Synaptic vesicle pools and active zone release apparatus in a presynaptic terminal. (**a**) Diagram represents synaptic vesicle (SV) localization in two pools in a presynaptic terminal. A recent in vitro study shows two distinct pools of tethered and reserved SVs. SVs coat on the surface of condensed liquid droplets (gray droplet) formed by active zone proteins RIM, RIM-BP, and ELKS via phase separation, that cluster Ca^2+^ channels anchored on lipid membrane bilayers. SVs coating on the surface of condensed liquid droplets connect the synapsin-clustered SV condensates (gray ellipse). Reproduced from Wu et al., 2021 [23]. The rectangle shows an SV-attached liquid droplet, and the protein components are shown in (**b**). (**b**) Diagram represents active zone proteins’ complex and a docked synaptic vesicle. The active zone is a highly organized structure that recruits voltage-gated Ca^2+^ channels and docks SVs close to fusion machinery proteins (SNAREs complex). Reproduced from Wang et al., 2016 [3]. The tight spatial organization ensure the fast exocytosis upon Ca^2+^ entry, and it provides molecular machinery to set and regulate synaptic strength, presynaptic short-term plasticity, and homeostatic synaptic plasticity.

**Table 1 ijms-22-11775-t001:** Active zone protein functions.

Function		Protein	References
AZ assembly	Ca_V_ channel recruitment	RIM, RIM-BP, CAST/ELKS	[35,37,38,39,42,43]
	liquid droplet formation	RIM, RIM-BP, ELKS	[23,48]
	stabilization and degradation	Bassoon, Piccolo	[51,52]
Fusion machinery interaction	fusion machinery regulation	Munc13, Munc18	[55,56,57,58,59,60]
SV states	tethering	Bassoon, Piccolo	[46,52,61]
	docking	RIM, CAST/ELKS	[3,10,62]
	priming	ELKS, RIM, RIM-BP, Munc13	[36,55,62,63,64,65]
	super priming	Mover	[22]
	fusion	Munc13, Munc18	[55,56,57,58,59,60]
SV replenishment	facilitation	Bassoon, Piccolo	[46,52,61]
	inhibition	CAST phosphorylation	[10]
Presynaptic short-term plasticity	facilitation	RIM, RIM-BP	[39,66,67]
	facilitation	Munc13, Munc18	[9,12,19,68]
	post-tetanic potentiation (PTP)	Munc18	[12,13]
	control of depression	Bassoon, Piccolo	[46,52]
	depression	CAST phosphorylation	[10]
Presynaptic homeostatic plasticity	RRP enlargement	RIM	[15]
	promotion of SV priming	RIM-BP	[69]
	promotion of SV replenishment	Bassoon	[16]

## Data Availability

Not applicable.

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
