# Peer review of "Stable and Flexible Synaptic Transmission Controlled by the Active Zone Protein Interactions"

_ijms, 2021, doi:10.3390/ijms222111775_

Round 1

Reviewer 1 Report

This is a very nice review. I only have suggestions for a few minor points (see the end of the review). The strengths and benefits of this review are:

1) It manages elegantly to cover both long-standing, well-established concepts fundamental for understanding active zone function, and in addition recently emerging models. This makes the review appealing both to experts and to scientists who begin to explore this field.

2) Among these novel concepts the review mentions:

  • reversible docking of SVs at active zones
  • a loose state and a tight state of docked SVs
  • data obtained by super-resolution imaging
  • active zone assembly by phase-separation

I think that these are indeed important and topical aspects in the context of active zone assembly and function and therefore good choices.

3) I like the layout and writing style. The author provides a story or narrative, starting with basic concepts and adding novel ideas successively, each of them well-explained. This allows the reader to remember the large number of facts and details obviously needed to understand this complex research topic. It also made reading this review a pleasant and inspiring experience.

I have the following minor points. Each of them is a suggestion, not a request:

a) The author mentions "loose states" in line 59. It may be helpful to add a few papers that introduce this term in line 60, where currently only reference 19 is cited. The "loose state - tight state" topic was introduced - as far as I know -  in Neher and Brose 2018 (DOI: 1016/j.neuron.2018.11.024) and Pofantis et al. 2021 (DOI: 10.1073/pnas.2022551118).

Reversible docking was also mentioned by He et al., 2017 (DOI: 10.1038/ncomms15915) and Chang et al., 2018 (DOI: 10.1038/s41593-017-0037-5). However, the focus of the paragraph should definitely remain on the Watanabe paper and the znap-and-freeze method. This is the well-chosen key paper for this paragraph. 

b) The author mentions Bassoon and Piccolo in the context of their direct roles for transmitter release. Bassoon has been implicated in stabilizing active zones by inhibiting their degradation. This could be mentioned briefly, e.g. in the paragraph on “AZ proteins and their complex”, because it adds another concept in addition to AZ assembly by phase-seperation. Papers could be, for example, Okerlund et al., 2017 (DOI: 1016/j.neuron.2017.01.026) and Waites et al., 2013 (doi: 10.1038/emboj.2013.27).

c) Maybe the title could be refined. "Stable and flexible synaptic transmission controlled by..." is a nice and original phrase. I am not sure what the second part means, i.e. "by the active zone proteins interaction". Maybe what is meant here is "by active zone protein interactions"? 

Author Response

Thank you very much for valuable comments. The correction is highlighted with blue in the text.

     a) The author mentions "loose states" in line 59. It may be helpful to add a few papers that introduce this term in line 60, where currently only reference 19 is cited. The "loose state - tight state" topic was introduced - as far as I know in Neher and Brose 2018 (DOI: 1016/j.neuron.2018.11.024) and Pofantis et al. 2021 (DOI: 10.1073/pnas.2022551118).

Thanks for the information. The "loose state - tight state" topic is described page 3 line 79-88.

      Reversible docking was also mentioned by He et al., 2017 (DOI: 10.1038/ncomms15915) and Chang et al., 2018 (DOI: 10.1038/s41593-017-0037-5). However, the focus of the paragraph should definitely remain on the Watanabe paper and the zap-and-freeze method. This is the well-chosen key paper for this paragraph. 

Thanks for the information. He et al, 2017 [61] is cited in Section 4, page 5, line 192-193.

     b) The author mentions Bassoon and Piccolo in the context of their direct roles for transmitter release. Bassoon has been implicated in stabilizing active zones by inhibiting their degradation. This could be mentioned briefly, e.g. in the paragraph on “AZ proteins and their complex”, because it adds another concept in addition to AZ assembly by phase-seperation. Papers could be, for example, Okerlund et al., 2017 (DOI: 1016/j.neuron.2017.01.026) and Waites et al., 2013 (doi: 10.1038/emboj.2013.27).

Thanks for the information. This topic is described in page 4 line 152- page 5 line 164.

     c) Maybe the title could be refined. "Stable and flexible synaptic transmission controlled by..." is a nice and original phrase. I am not sure what the second part means, i.e. "by the active zone proteins interaction". Maybe what is meant here is "by active zone protein interactions"? 

Thanks for the comment. The title is corrected.

Reviewer 2 Report

The article provides a very detailed and interesting review of specialized protein machinery responsible for the release and recycling of neurotransmitter vesicles in the central synapses. Potentially, the paper would be interesting for broad audience, in particular the neuroscientists working in the area of synaptic transmission and plasticity. In my opinion, paper could by published almost as it is, with few minor clarifications.

  1. The section 2 (2nd paragraph) discusses interesting experiments on temporal dynamics of vesicle fusion. It would be great if this section was accompanied by figure, illustrating the timeline of changes in the AZ and different pools of vesicles. Also, accordingly the data obtained by the ‘zap-and-freeze’ method, main events of fusion occur within 5-11 ms after stimulated. This contrasts with the temporal parameters of evoked synaptic currents – delay < 1 ms and typical rise time of 1-3 ms. Could Author comment on this?

  1. The review repeatedly mentions the essential protein components of AZ, such as Mucn13, RIM, RIM-BP, Basoon, etc. in different sections dedicated to the different aspect of synaptic function, such as SV replenishment, pre-synaptic plasticity. It would be beneficial for perspective readers, especially students, if Author could include a table to summarize main functions of these proteins and outline their putative physiological and pathological implications.

    3. Page 3, lines 123-126: the sentences, staring from “ Remark-

ably, SUV-coated…” and “ The SUV-coated “ should contain references, otherwise it is not clear what papers are mentioned in the next sentence, starting from “These studies reconstitute….”.

  1. Page 4, last paragraph (“SV fusion”): the last two sentences seem to contradict each other – if synaptic connectivity is dependent on neurotransmitter secretion, how could deletion of Munc13 come not to prevent normal synaptic assembly. Also, “brain assembly…” (line 175) sounds confusing, “normal synaptic assembly and formation of morphologically-defined synapses” would be better. Also, the work cited ([63]), is rather out-dated (2000). Could Author discuss more recent studies on this topic ?

  1. In general, the paper would benefit from the careful check of the language and writing style and correcting minor stylistic errors and streamlining some long and complex sentences, which are difficult to follow sometimes. Some examples and suggested amendments are given below:

  • page 3, lines 81-82: the first sentence should be modified like “ The AZ is highly-organized multi-protein structure serving as a platform for SV exocytosis mediated by SNARE complex and nearby CaV channels”
  • 3, line 85 “Ca2+-sensor”
  • 3, lines 92-93: The sentence starting from “Thus, RIM deletion reduces…[35]” should continue “…. impairs SV docking and slows down the rate of exocytosis”.
  • 3, line 122– the expression “coat on”, which is used here and so forth, sounds jargonish and may confuse readers outside of specific research area. It should be replaced by “forms the coat on” or “coats [some surface] …” or “assemble on …”
  • 4, line 145 : the comma is needed after “As described above”
  • 6, lines 244-246: This sentence is confusion and might be modified as “Short-term synaptic plasticity occurs during and after repetitive presynaptic AP activity on a time scale of milliseconds to minutes and manifests in the modulation of synaptic efficacy via changes in the release probability and RRP size”
  • 7 line 287: “These findings indicate dependent and independent mechanisms… “ is confusion, did Author mean that Munc18 phosphorylation can have different effects in different synapses ?

Author Response

Thank you very much for valuable comments. The correction is highlighted with yellow in the text.

1. The section 2 (2nd paragraph) discusses interesting experiments on temporal dynamics of vesicle fusion. It would be great if this section was accompanied by figure, illustrating the timeline of changes in the AZ and different pools of vesicles. Also, accordingly the data obtained by the ‘zap-and-freeze’ method, main events of fusion occur within 5-11 ms after stimulated. This contrasts with the temporal parameters of evoked synaptic currents – delay < 1 ms and typical rise time of 1-3 ms. Could Author comment on this? 

Thanks for the comment. A scheme is added to page 6. It explains the main events of fusion occur within 5 ms. Comparison with evoked synaptic currents is discussed page 3 line 76-78.

2. The review repeatedly mentions the essential protein components of AZ, such as Mucn13, RIM, RIM-BP, Basoon, etc. in different sections dedicated to the different aspect of synaptic function, such as SV replenishment, pre-synaptic plasticity. It would be beneficial for perspective readers, especially students, if Author could include a table to summarize main functions of these proteins and outline their putative physiological and pathological implications.

Thanks for the comment. A table is added in page 6, but pathological implications cannot be included because I am not expert of the field.

3.  Page 3, lines 123-126: the sentences, staring from “ Remark-ably, SUV-coated…” and “ The SUV-coated “ should contain references, otherwise it is not clear what papers are mentioned in the next sentence, starting from “These studies reconstitute….”. 

 Reference [25] is added.

4. Page 4, last paragraph (“SV fusion”): the last two sentences seem to contradict each other – if synaptic connectivity is dependent on neurotransmitter secretion, how could deletion of Munc13 come not to prevent normal synaptic assembly. Also, “brain assembly…” (line 175) sounds confusing, “normal synaptic assembly and formation of morphologically-defined synapses” would be better. Also, the work cited ([63]), is rather out-dated (2000). Could Author discuss more recent studies on this topic ? 

   ‘dependent’ in the last sentence is ‘independent’. So, two sentences were contradictory. The typo is corrected.

The confusing sentence is corrected following reviewer’s comment (page 5 line 209). And recent works are discussed in page 6 line 210-215.   

5. In general, the paper would benefit from the careful check of the language and writing style and correcting minor stylistic errors and streamlining some long and complex sentences, which are difficult to follow sometimes. Some examples and suggested amendments are given below: 

  • page 3, lines 81-82: the first sentence should be modified like “The AZ is highly-organized multi-protein structure serving as a platform for SV exocytosis mediated by SNARE complex and nearby CaV channels”.   

              The sentence is corrected as the reviewer’s suggestion.

  • 3, line 85 “Ca2+-sensor”.     

              Corrected.

  • 3, lines 92-93: The sentence starting from “Thus, RIM deletion reduces…[35]” should continue “…. impairs SV docking and slows down the rate of exocytosis”.

              The sentence is corrected as the reviewer’s suggestion.

  • 3, line 122– the expression “coat on”, which is used here and so forth, sounds jargonish and may confuse readers outside of specific research area. It should be replaced by “forms the coat on” or “coats [some surface] …” or “assemble on …”.                                  

              The sentence is corrected as the reviewer’s suggestion.

  • 4, line 145 : the comma is needed after “As described above”

              The comma is added.

  • 6, lines 244-246: This sentence is confusion and might be modified as “Short-term synaptic plasticity occurs during and after repetitive presynaptic AP activity on a time scale of milliseconds to minutes and manifests in the modulation of synaptic efficacy via changes in the release probability and RRP size”.

                The sentence is corrected as the reviewer’s suggestion.

  • 7 line 287: “These findings indicate dependent and independent mechanisms… “ is confusion, did Author mean that Munc18 phosphorylation can have different effects in different synapses ?

                The confusing sentence is corrected.
